# Examining Determinants of Corruption at the Individual Level in South Asia

**Jinwon Han** [1,2,3]

1. Graduate School of International and Area Studies (GSIAS), Hankuk University of Foreign Studies, Seoul 02450, Republic of Korea; jinwonhan@hufs.ac.kr
2. Center for International Cooperation & Strategy (CICS), Hankuk University of Foreign Studies, Seoul 02450, Republic of Korea
3. Gachon Liberal Arts College, Gachon University, Seongnam 13120, Republic of Korea

**Abstract:** Although the topic of corruption in South Asia has attracted a great deal of attention, extant research on individual-level factors is sparse. With this background, this paper examines the effect of selected individual-level determinants on South Asian people's justifiability of corruption. For analysis, the paper uses the World Value Survey (WVS) dataset and carries out its own survey as well. Using multinomial logistic regression (MLR) and binomial logistic regression (BLR) as a robustness check, this paper identifies that variables of age, education, religiosity, and individualism/collectivism have significant effects on respondents' corruption justifiability in South Asia. In addition, the paper performs additional analyses independently using two surveys utilized in the main analysis as a single dataset to report the main differences between them. In conclusion, this paper proposes a region-specific suggestion for South Asian governments and policy makers.

**Keywords:** corruption; individual-level determinants of corruption; South Asia; multinomial logistic regression (MLR); binomial logistic regression (BLR)





## 1. Introduction

Persistent corruption[1] has been a long-standing challenge in South Asia. With few exceptions, most South Asian countries have ranked among the most corrupt countries in the world. For example, apart from Bhutan, the remaining seven countries (i.e., Afghanistan, Bangladesh, India, Nepal, Maldives, Pakistan, and Sri Lanka) received under 50 points out of 100 in Transparency International's 2022 Corruption Perceptions Index (CPI) (Transparency International (TI 2023)).[2] Given that South Asia has become one of the most dynamic and attractive emerging economies in recent decades with its high economic growth compared to other areas (Bhattacharyay 2014) and is now a major global hub of manufacturing and trade for the world economy, the adverse effects of corruption in the region will not be confined to the region itself but will spread worldwide. In this context, investigating determinants of corruption in the context of the South Asian region is an important and timely matter. In response to this pressing issue, many scholars have explored several determinants of corruption to address the issue in the region (see Han 2022).[3]

Nonetheless, a considerable proportion of the previous research has been primarily limited to the national level, neglecting the critical role of individual-level factors in perpetuating corruption. As corrupt practices are ultimately committed by individuals, identifying the determinants of corruption at the individual level is just as crucial as investigating country-level determinants. However, extant research focusing on individual-level factors has been sparse (Hunady 2017).

The dearth of research on individual-level determinants of corruption is primarily attributed to the earlier studies that centered on individuals' backgrounds and personalities to account for their corrupt behavior. While these studies established a causal chain from

an individual's flawed character to their engagement in corrupt practices (De Graaf 2007), they lost their momentum in the 1980s due to the lack of empirical evidence to support their claims.

In recent times, however, a few studies investigating individual-level determinants of corruption have shifted their focus from a person's unique personality traits to universal factors at the individual level, such as age, gender, and so forth. This alternative approach has several advantages, including facilitating cross-national research on individual-level determinants of corruption, which can offer insight into where anti-corruption policy should best be targeted.

While emerging scholarship has been providing insight into several plausible individual-level factors affecting corruption as such, no previous studies, to the best of the author's knowledge and based on searches of peer-reviewed databases, have sought to answer the crucial research question, "then, among those individual-level factors identified in the existing literature, which determinant(s) has a significant effect on corruption in South Asia?" Even though certain factors are revealed as significant either globally or elsewhere, their effects and significance would vary depending on the regional context. Hence, it is crucial to re-examine the impact of those revealed individual-level determinants of corruption specifically in the context of the South Asian region.

With this background, this paper aims to investigate the impact of individual-level factors on corruption in South Asia[4], thereby seeking to address this research gap. More specifically, it will examine the effect of selected individual-level determinants on corruption in six South Asian countries, where all the data are available. The results of this paper will be expected to extend the limited research on the understanding of individual-level factors and their impacts on corruption in the context of the South Asian region. For analysis, multinomial logistic regression (MLR) and binomial logistic regression (BLR) as a robustness check are utilized. The paper will elaborate on these methodologies in Section 3.

The subsequent sections of this paper are organized as follows. In Section 2, this paper reviews an extensive set of previous works on individual-level determinants of corruption to select pertinent determinants for analysis. Section 3 provides a description of the data and methodologies. In Section 4, this paper carries out empirical analyses and a robustness check and presents a discussion of the results and their implications. Lastly, Section 5 concludes with the academic contributions of this paper and policy implications for South Asian governments and decision makers.

## 2. Literature Review

An in-depth examination of prior studies indicates that the following individual-level factors have a significant association with corruption: (1) age, (2) gender, (3) marital status, (4) education, (5) religion, (6) trust, and (7) individualism/collectivism.

### 2.1. Age

Prior research on age differences is based on the theoretical assumption that, all other factors being equal, individuals have an increased likelihood of interacting with public officials as they age. Consequently, the probability of exposure to corrupt practices and the corresponding corruption justifiability are likely higher as individuals advance in age. Nevertheless, after surpassing a specific age threshold, they become less tolerant of corruption, resulting in a decreased likelihood of engaging in corrupt practices.

Through an analysis of micro-datasets, Swamy et al. (2001) discovered that the age variable exhibits a negative correlation with individuals' tolerance for corruption, indicating that as individuals age, they are less likely to tolerate corrupt practices. Given that individuals' justifiability of corruption is closely associated with their actual involvement in corrupt activities (Ajzen and Fishbein 1980) and statistically correlated with established corruption indices (Torgler and Valev 2006), it is reasonable to assert that one's corruption justifiability can be a proxy for their engagement in corruption (Han 2022).[5]

Using a more recent dataset collected in the mid-1990s, Torgler and Valev (2006) expanded on Swamy and colleagues' research. The authors found that all age groups from 30 to 65+ report significantly lower corruption justifiability than the reference group below 30. This finding indicates that people are less likely to rationalize corrupt practices as they age. The same authors obtained a similar result from their analysis conducted in 2010 using survey data from eight Western European countries (Torgler and Valev 2010).

In a similar vein, Lavena (2013) examined the effects of various individual factors, including age differences, on people's corruption permissiveness (justifiability) in six Latin American countries. The analysis demonstrated that age was negatively associated with corruption permissiveness, indicating that younger people are more likely to justify corruption, while older people are less likely to accept it. This result was consistent across all models and demonstrated strong robustness.

Hunady (2017) investigated the influence of age on respondents' experiences of being victims of corruption and attitudes toward corruption. Regarding experiences of corruption, the author discovered that individuals up to the age of 34 are more prone to be victims of corruption, whereas the likelihood of experiencing corruption tends to decline after the age of 34. On the other hand, the analysis of corruption tolerance demonstrated that respondents are less inclined to accept corruption as they age. Based on these arguments, this paper derives the following hypothesis.

**H1:** *Relatively more younger people are more likely to perceive corruption as acceptable, whereas relatively more older people are less likely to accept it.*

### 2.2. Gender

In addition to age differences, numerous studies have investigated the relationship between gender and corruption. Researchers have posited that, holding all else constant, males are typically more susceptible to corruption than females due to their greater likelihood of being targeted for corrupt activities and their higher levels of tolerance for such behavior (Mocan 2008).

Consistent results have been obtained by scholars such as Swamy et al. (2001), Torgler and Valev (2006), Mocan (2008), Torgler and Valev (2010), and Hunady (2017). In contrast, Lavena (2013) found the contrary result that gender differences are insignificant to Latin American respondents' tolerance toward corruption.

In contrast to the aforementioned scholars, Alatas et al. (2009) devised a one-shot experimental bribery game that featured three roles, namely, a firm, a government official, and a citizen. Their investigation revealed that men were more inclined to offer and accept bribes than women in the sample of a Western country. However, no statistically significant gender differences were observed in the other three Asian samples.

Frank et al. (2011) conducted a comprehensive review of various experimental studies on corruption to examine the impact of gender on the propensity to offer and accept bribes. After analyzing six studies, the authors noted a consistent pattern of results, suggesting that being female is associated with less exposure and/or lower tolerance toward corrupt practices.

Rivas (2013) employed a more refined approach by conducting a laboratory-based bribery experiment. The experimental design involved a repeated two-person game spanning 20 rounds, featuring two roles, namely, a firm and a public officer. The findings of this investigation demonstrated that when women played the role of a firm, they were less inclined to offer bribes than men. Moreover, even when women did offer bribes, the amount was substantially lower than the average amount offered by men. Furthermore, the frequency with which women accepted bribes when playing the role of a public officer was lower than that of men, and even when they did accept a bribe, women were less likely to engage in corrupt practices than men. Based on these arguments, this paper derives the following hypothesis.

**H2:** *Males are more likely to perceive corruption as acceptable, whereas females are less likely to accept it.*

*2.3. Marital Status*

In addition, many studies have examined the relationship between individuals' marital status and their propensity toward engaging in corrupt activities. Scholars in this field have asserted that marital status is a significant factor of corruption based on the life-course theory. This theory posits that marriage is a critical juncture in an individual's life, which can influence their public behavior (Swamy et al. 2001), as well as their ability to comprehend information and adhere to rules (Melgar et al. 2010). Thus, marriage can play a pivotal role in shaping an individual's tolerance toward corruption, thereby impacting their likelihood of engaging in corrupt practices.

For instance, Swamy et al. (2001) and Torgler and Valev (2006) found that married individuals are less inclined to tolerate corruption. Specifically, Torgler and Valev (2006) observed a significant correlation between the marital status of individuals and their justifiability of corruption across all the regions they surveyed, ranging from Western Europe to Africa. In a subsequent study conducted in 2010, the same authors confirmed that being married is significantly associated with a lower level of the justifiability of corruption, even after accounting for country-specific heterogeneity. Based on these arguments, this paper derives the following hypothesis.

**H3:** *Unmarried people (e.g., single, divorced, and otherwise) are more likely to perceive corruption as acceptable, whereas married people are less likely to accept it.*

*2.4. Education*

Scholars have also emphasized the role of individual education levels in shaping their perception of corruption and propensity to engage in corrupt behaviors. However, while these studies have substantially contributed to the field of corruption research, most have not provided a clear theoretical framework for the link between education and corruption.

Nonetheless, we can see a hint of the causal mechanism through which individuals' education levels affect their perceptions and behaviors from Uslaner and Rothstein (2016). According to the authors, a high level of educational attainment matters for controlling corruption at the individual level because as people become more educated, they can be better at reading, establishing social bonds with different communities, having a sense of citizenship and loyalty toward the state, and complaining more about corruption, all of which are negatively associated with engagement in corruption.

Swamy et al. (2001) introduced a dummy variable for education as a control and found a negative correlation between the education variable and individuals' corruption justifiability. This result supports the theoretical expectation that those with higher education are less likely to tolerate corrupt behaviors.

In addition to Swamy et al. (2001), Truex (2011) and Hunady (2017) also identified a significant negative association between individuals' education levels and their tolerance toward corruption. Truex's (2011) study is particularly noteworthy, as it focuses on the Nepalese context, one of the South Asian countries being covered in this paper. The study found that higher education levels are significantly associated with lower justifiability of corruption in the country.

Consistent with Truex (2011), Hunady (2017) also obtained a similar result. The study found a significant positive association between respondents' education levels and their experience of corruption, as well as a negative correlation between individuals' education levels and their tolerance of corruption. These findings suggest that although more educated people are more likely to be at risk of becoming victims of corruption, they are less likely to tolerate it. Based on these arguments, this paper derives the following hypothesis.

**H4:** *Relatively less educated people are more likely to perceive corruption as acceptable, whereas relatively more educated people are less likely to accept it.*

*2.5. Religion*

Previous studies have also highlighted the significance of religion in relation to corruption.[6] Nevertheless, it is essential to distinguish between two distinct strands of this inquiry, one that examines the influence of individuals' religious affiliations on the justifiability of corruption and the other that explores the impact of their level of religiosity (Gokcekus and Ekici 2020).

The theoretical basis for the first branch of this religious study was mainly drawn from La Porta et al. (1996) and Treisman (2000). These researchers highlighted the role of religion in shaping people's conduct by contending that a person's religious affiliation affects their perceived costs of engaging in corrupt activities, whether it is the fear of being caught or the fear of punishment. They further suggested that in regions where more hierarchical religions, such as Catholicism, Eastern Orthodoxy, and Islam, are prevalent, people are more likely to justify corruption than those in regions dominated by egalitarian or individualistic religions, such as Protestantism.

In a similar vein, Paldam (2001) conducted a comprehensive analysis of the influence of various religious traditions on corruption. Categorizing Christianity into pre-reformation Christians (i.e., Catholic and Eastern Orthodox) and reform Christians (i.e., Protestant and Anglican), the author reported that reform Christianity and Tribal religion reduce corruption, while the remaining religions increase it.

Gerring and Thacker (2004) documented a negative correlation between Protestantism and corruption across all models. Consistent with Gerring and Thacker's findings, You and Khagram (2005) found a negative link between Protestantism and corruption in all their estimations. Additionally, this relationship was upheld in further analysis using instrumental variables, albeit with a reduced level of significance.

Sommer et al. (2013) noted a significant and negative correlation between Protestantism/Confucianism and corruption, indicating that nations with a legacy of Protestantism or Confucianism tend to have lower levels of corruption. Conversely, Hinduism was found to be positively associated with corruption, suggesting that countries in which Hinduism is predominant are more prone to corruption. However, none of these associations were significant in the individual-level analysis. Instead, Catholicism and Orthodoxy emerged as the key factors associated with an increased tolerance for corruption.

The second strand of research on religion and corruption shifted its focus from religious affiliation to individual religiosity. Scholars in this strand critically reviewed the earlier literature and argued that belonging to a religious group does not necessarily imply full obedience and practice of the teachings of that group (Gokcekus and Ekici 2020). Instead, they suggested that the level of devotion to religion (religiosity) may be more directly and negatively associated with corruption than one's religious affiliation.

Drawing on this theoretical expectation, Yeganeh and Sauers (2013) and Gokcekus and Ekici (2020) examined the effects of religiosity on corruption. In their analyses, they similarly found an insignificant relationship between religious denominations and corruption as anticipated, but also an unexpected positive correlation between religiosity and corruption.

In Zakaria (2018), a similar noteworthy finding was observed. The author measured religiosity using two proxies: the frequency of attending religious services and private prayer outside of religious services. The analysis revealed a significant negative correlation between the frequency of private prayer and tolerance of corruption, indicating that individuals who pray less frequently are more likely to tolerate corruption. Meanwhile, the author found a positive correlation between the frequency of attendance at religious services and tolerance of corruption. Despite this unexpected finding, the author did not offer an in-depth explanation as to why this correlation is established. Based on these arguments, this paper derives the following hypotheses.

**H5:** *People having relatively more hierarchical religions (e.g., Catholicism, Eastern Orthodoxy, and Islam) are more likely to perceive corruption as acceptable, whereas people having relatively more egalitarian religions (e.g., Protestantism and so on) are less likely to accept it.*

**H6:** *People who attend religious services more frequently are more likely to perceive corruption as acceptable, whereas people who attend religious services less frequently are less likely to accept it.*

**H7:** *People who pray less frequently are more likely to perceive corruption as acceptable, whereas people who pray more frequently are less likely to accept it.*

*2.6. Trust*

Several studies have examined the impact of trust on corruption as well. This strand of literature is grounded in the theoretical perspective that corruption is a type of collective action problem (Persson et al. 2013; You 2017).

Suppose that in a certain society, corruption is the expected behavior. In such a circumstance, even though people condemn corrupt behaviors or recognize that they would be better off jointly refraining from corruption, they have no reason to cooperate to minimize it because there are no benefits but rather substantial costs of acting fairly. Hence, people will likely choose corrupt alternatives rather than non-corrupt ones in this setting (Persson et al. 2013).

Here, it is important to note that such a cost–benefit calculation is ultimately derived from the individual (dis)trust in the truthfulness of other people and the existing monitoring institutions. More specifically, people are likely to rationalize their own corrupt behaviors in thoroughly corrupt settings because they cannot easily trust that most others and the existing institutions will act fairly. However, if individuals trust that others will refrain from corrupt practices and that monitoring institutions will be fair, they are more likely to resist corruption to pursue the long-term benefits of impartiality (You 2017).

Uslaner (2004) was the first attempt to investigate an association between trust and corruption.[7] The author analyzed the causal impact of trust on corruption and vice versa. The results revealed that trust has a greater effect on corruption than the reverse causal relationship. Drawing from these findings, the author asserted that the causal direction between trust and corruption runs from trust to corruption.

Consistent with Uslaner's findings, Bjørnskov (2003) also observed a negative correlation between trust and corruption. This result implies that a society with a high level of trust tends to exhibit low levels of corruption. This finding was so strong that it was robust to the inclusion of multiple confounding variables.

Chang (2012) examined the relationship between trust and corruption in a more nuanced manner by differentiating between two types of trust: (1) generalized trust and (2) institutional trust. Generalized trust refers to an individual's trust in other members of society, whereas institutional trust pertains to one's trust in a country's institutions and civil services. The analysis revealed that generalized trust was significantly and negatively associated with corruption, suggesting that societies in which individuals have higher levels of trust in one another are less likely to experience corruption. In contrast, institutional trust was found to have no significant relationship with corruption.

Kubbe (2013) examined the impacts of interpersonal (generalized) and institutional trusts on corruption. However, the author took a slightly different approach by formulating a hypothetical causal model, where corruption mediates the relationship between interpersonal trust and institutional trust. This model posits that interpersonal trust influences corruption, and, in turn, corruption influences institutional trust. The analysis demonstrated that interpersonal trust is negatively associated with corruption and, as expected, corruption is also negatively associated with institutional trust. Based on these arguments, this paper derives the following hypotheses.

**H8:** *People who portray themselves as less trusting in other people are more likely to perceive corruption as acceptable, whereas people who portray themselves as more trusting in other people are less likely to accept it.*

**H9:** *People who portray themselves as less trusting in a country's institutions and civil services are more likely to perceive corruption as acceptable, whereas people who portray themselves as more trusting in a country's institutions and civil services are less likely to accept it.*

### 2.7. Individualism/Collectivism

Lastly, several scholars focused on the role of two cultural values, individualism and collectivism, in explaining corruption. According to their theoretical predictions, societies with a more individualistic culture or people with more individualistic characteristics are less likely to promote or engage in corruption, while societies with a more collectivistic culture or people with more collectivistic traits are more likely to foster or engage in corruption.

These studies are based on a theoretical framework that people in more individualistic cultures or with individualistic characteristics tend to view themselves as autonomous entities and value personal achievements over in-group goals, resulting in less engagement in corruption. In contrast, people in more collectivistic cultures, where individuals emphasize cohesiveness and prioritize the group over the self, or with collectivistic traits, are more likely to prioritize relationships over rules, which causes several forms of corruption, such as favoritism and nepotism (Zhang 2020).

Davis and Ruhe (2003) posited that individualism is associated with lower corruption, while collectivism is associated with higher corruption. Their empirical analysis supported this hypothesis by demonstrating that individualism is significantly and negatively related to corruption, while collectivism is significantly and positively related to corruption. Moreover, this relationship remained significant even after controlling for other cultural variables, although the significance level decreased.

Jha and Panda (2017) also investigated the effects of individualism/collectivism on corruption. The authors hypothesized that in individualistic societies, where individual autonomy and achievements are more valued, people may feel more responsible for their behaviors and hence are less inclined to engage in corruption. In contrast, in more collectivistic societies, people may feel obligated to distribute rewards more generously to their group members, resulting in favoritism or nepotism. The authors' analysis revealed that individualism and collectivism are indeed negatively and positively associated with corruption, respectively, in all models. Furthermore, this finding remained robust after controlling for a set of variables and conducting several sensitivity analyses.

On the contrary, a few scholars claimed an alternative perspective to the aforementioned studies. One such scholar is Suh (2018), who argues that individualism, with its emphasis on personal meritocracy and success, may lead individuals to behave opportunistically in pursuit of their desired rewards. As a result, individualists may be more inclined to rationalize engaging in corrupt behavior as a means of achieving their goals. The author's empirical analysis also proved a positive association between individualism and corruption. Based on these arguments, this paper derives two opposite hypotheses.

**H10:** *People with more collectivistic characteristics are more likely to perceive corruption as acceptable, whereas people with more individualistic characteristics are less likely to accept it.*

**H10′:** *People with more individualistic characteristics are more likely to perceive corruption as acceptable, whereas people with more collectivistic characteristics are less likely to accept it.*

## 3. Data and Method

For the empirical analysis, this paper uses the degree of corruption tolerance among respondents reported in the seventh wave of the World Value Survey (WVS) 2017–2020 dataset as a proxy for the outcome of interest. The WVS dataset captures civic attitudes and opinions on a range of issues across a broad set of countries. The dataset's extensive coverage and representative sample size have made it a reliable choice for scholars and comparativists.

In addition to its comparative advantages, this paper employs the WVS data for the following methodological reasons. Firstly, since multiple independent variables at the individual level are used in this paper, it is more appropriate to have the dependent variable correspondingly at the individual level. Secondly, as noted by many scholars, individuals' actual involvement in corrupt activities is closely related to the extent to which

they perceive corruption as justifiable (Torgler and Valev 2006; Hunady 2017). Thirdly, experience-based data, which is an alternative measurement of corruption at the individual level, is susceptible to under-reporting issues, as respondents may be unwilling to report their own misbehaviors.

The WVS assesses participants' tolerance toward corruption by inquiring, "please tell me for the following statement whether you think it can always be justified, never be justified, or something in between: (…) someone accepting a bribe in the course of their duties". The values range from 1 for "never justifiable" to 10 for "always justifiable". Following Torgler and Valev (2010), this paper recodes this index on a scale of 0 to 3, where 0 indicates "always justifiable," 1 indicates "somewhat more justifiable," 2 indicates "somewhat less justifiable," and 3 indicates "never justifiable" for simplicity and explicitness.

To operationalize the selected independent variables, this paper uses the same WVS dataset. Individual age differences are measured using a 3-point scale ranging from 1 to 3, where 1 represents ages up to 29, 2 represents ages 30–49, and 3 represents ages 50 and more. Gender differences are measured by a 2-point scale ranging from 1 to 2, where 1 represents males and 2 represents females. This paper recodes it into a binary variable with a value of 1 for males and 0 for females. Respondents' marital status is rated on a 6-point scale in the dataset, with 1 representing married, 2 representing living together as married, 3 representing divorced, 4 representing separated, 5 representing widowed, and 6 representing single. This paper recodes the original 6-point scale into a 2-point scale, with a value of 1 for being married and 0 otherwise.

Next, the WVS measures respondents' education levels using a 9-point scale based on the International Standard Classification of Education Degrees (ISCED) 2011. The scale ranges from 0 for early childhood education (ISCED 0)/no education to 8 for doctoral or equivalent (ISCED 8), with intermediate categories for primary education (ISCED 1), lower secondary education (ISCED 2), upper secondary education (ISCED 3), post-secondary non-tertiary education (ISCED 4), short-cycle tertiary education (ISCED 5), bachelor or equivalent (ISCED 6), and master or equivalent (ISCED 7). This paper recodes the original 9-point scale into a 3-point scale with low (ISCED 0–2), medium (ISCED 3–4), and high (ISCED 5 or higher) educational levels.

This paper divides the religious factor into two sub-variables, i.e., individuals' religious denominations and religiosity, based on prior research. To assess various religious affiliations, this paper employs the WVS's item inquiring about respondents' religious denominations.[8] To gauge religiosity, the paper utilizes two items about the frequency of attending religious services and prayer, following Zakaria (2018). The first item is initially rated on a 7-point scale, where 1 corresponds to frequent attendance and 7 corresponds to non-attendance. Similarly, the second item queries respondents about the frequency of their prayer and measures it on an 8-point scale, ranging from a score of 1 if the respondent frequently prays to God and 8 if the respondent never prays. To facilitate the interpretation, this paper reversely recodes both items so that higher scores indicate frequent attendance and prayer.[9]

Regarding the variable of trust, this paper distinguishes it into two kinds of trust, generalized trust (also called interpersonal trust) and institutional trust, following the existing literature. To measure generalized trust, this paper utilizes a frequently used item from the WVS that asks respondents, "generally speaking, would you say that most people can be trusted or that you need to be very careful in dealing with people?" This item is measured on a binary scale, where 1 denotes "most people can be trusted" and 2 denotes "need to be very careful". To measure institutional trust, this paper employs the item designed to capture respondents' confidence in the civil service, following Chang (2012). A level of respondents' confidence in civil service is measured on a 4-point scale, where a value of 1 indicates high confidence and 4 indicates low confidence. For ease of interpretation, this paper recodes both items' scores in a reverse way, with higher values indicating higher trust.

To operationalize the individualism/collectivism construct, this paper uses two items from the WVS that inquire about respondents' life satisfaction and freedom of choice and

control, which have a strong correlation with Hofstede's individualism/collectivism index (Kang and Kwon 2018).[10] Respondents' life satisfaction is measured by the WVS using a 10-point scale ranging from 1 indicating complete dissatisfaction to 10 indicating complete satisfaction. Similarly, respondents' perception of the level of freedom of choice and control over their lives is assessed using a 10-point scale, with a value of 1 indicating no choice at all and 10 indicating a great deal of choice. As both items measure respondents' perceptions in order, this paper employs both variables without recoding. The detailed descriptions of each variable and their recoding are summarized in Table 1.

**Table 1.** Variable descriptions.

| Variables | Descriptions | Recoded Range |
| --- | --- | --- |
| Corruption Justifiability | Respondents' tolerance toward corruption | 0~3 |
| Age | Respondents' age in years | 1~3 |
| Gender | Respondents' gender | 0~1 |
| Marital Status | Respondents' marital status | 0~1 |
| Education | Respondents' educational attainments | 1~3 |
| Religious Denomination | Respondents' self-reported religious affiliations | 0~9 |
| Attendance (Religiosity) | The frequency of respondents' attendance at religious services | 1~7 |
| Prayer (Religiosity) | The frequency of respondents' prayer | 1~8 |
| Generalized Trust | The level of respondents' trust in people in a country | 1~2 |
| Institutional Trust | The level of respondents' trust in institutions and civil services in a country | 1~4 |
| Life Satisfaction (Individualism/collectivism) | The level of respondents' life satisfaction | 1~10 |
| Freedom of Choice and Control (Individualism/collectivism) | The level of respondents' perceived freedom of choice and control over their lives | 1~10 |

Despite its comparative merits, the seventh wave of the WVS dataset is inadequate for this paper since it only encompasses respondents from Bangladesh and Pakistan, but not the remaining four South Asian countries, Bhutan, India, Nepal, and Sri Lanka. To address this data limitation, this paper conducted its own survey to obtain responses from these four countries.

In this survey research, this paper replicates the WVS's original questionnaire written in English for comparability and translates it into the respective countries' official languages, following Han (2022).[11] The translation of the survey questionnaire was conducted by native speakers from each South Asian country who possess a doctoral degree, or at least are doctoral candidates, or by a professional individual who has an excellent command of the language.[12]

Afterward, this paper electronically distributed a survey questionnaire to the general population of the respective countries without any quotas using an online survey tool provided by Google, namely Google Forms. While face-to-face data collection is generally considered the most optimal and reliable method for survey research, this approach was not feasible due to lockdowns in place at the time of the survey and the need to mitigate the risk of COVID-19 transmission.

The survey was initiated on 25 January 2022 and was closed on 6 March 2022, with a total of more than 1400 responses (N = 1426) recorded. In this online survey research, each question was mandatory, and only one response per participant was allowed to avoid incorrect answers. Nonetheless, one response from Bhutan was excluded from the analysis due to incorrect recording. After removing incomplete responses, the total sample size was 4190 individuals, including the original samples from the WVS.[13] This sample size meets

the recommended guideline for MLR, which is a minimum of 10 cases per independent variable (Schwab 2002).

This paper employs an MLR model to ascertain the individual variables' relative contributions to people's tolerance for corruption in South Asia. Logistic regression is devised to estimate the probability of the occurrence of an event by fitting data to a logistic curve. It is typically used when the dependent variable is categorical (mostly binary variables) and corresponding independent variables are either categorical or continuous. Among different types of logistic regression, this paper particularly uses an MLR model because our dependent variable is not restricted to two categories, and the assumptions of proportional odds (PO) and parallel line (PL) for ordinal probit regression (OPR) and logistic regression (OLR) models, respectively, do not hold in this paper ($p < 0.001$). To address this issue, García-Pérez (2013) suggests employing an MLR model as an alternative when dealing with ordinal variables.

To empirically assess the goodness-of-fit of the MLR model, this paper performs the Likelihood ratio test, which compares the complete model to the intercept-only model (i.e., no predictors).[14] The result of this test indicates that the full model containing all the predictors in this paper shows a significant improvement in fit over the null model with no predictors ($p < 0.001$). With this background, this paper uses an MLR model as the main analytical technique. Table 2 presents the result of the goodness-of-fit test.

**Table 2.** Goodness-of-fit and likelihood ratio tests.

| Measure | Model | Model Fitting Criteria | Likelihood Ratio Tests | | |
|---|---|---|---|---|---|
| | | −2 Log Likelihood | Chi-Square | df | *p*-Value |
| Model Fitting Information | Final | 7582.774 | 382.484 | 57 | <0.001 |

An econometric model is as follows:

$$Y = \ln(P_j/P_k) = \beta_{j0} + \beta_{j1}x_{j1} + \beta_{j2}x_{j2} + \beta_{j3}x_{j3} + \beta_{j4}x_{j4} + \beta_{j5}x_{j5} + \beta_{j6}x_{j6} \\ + \beta_{j7}x_{j7} + \beta_{j8}x_{j8} + \beta_{j9}x_{j9} + \beta_{j10}x_{j10} + \beta_{j11}x_{j11} \tag{1}$$

where:

$Y$ = the probability that an individual is in a particular corruption justifiability category $j$ divided by the probability of being in the reference category $k$;
$\beta_0$ = the constant;
$\beta_{1,2,3,...,k}$ = the coefficient of the selected independent variables;
$x_1$ = age differences;
$x_2$ = gender differences;
$x_3$ = marital status;
$x_4$ = a level of education;
$x_5$ = religious denominations;
$x_6$ = frequency of attendance at religious services (as a measure of religiosity);
$x_7$ = frequency of prayer (as a measure of religiosity);
$x_8$ = a level of generalized trust;
$x_9$ = a level of institutional trust;
$x_{10}$ = a level of life satisfaction (as a measure of individualism/collectivism);
$x_{11}$ = a level of freedom of choice and control (as a measure of individualism/collectivism).

## 4. Analysis and Results

Among the respondents, 36.4 percent are aged up to 29, 49.9 percent are between 30 and 49 years old, and 13.7 percent are aged 50 and above. The majority of participants are married (72.2%), while 27.8 percent are otherwise. The gender distribution is slightly skewed toward males, with 54.7 percent male and 45.3 percent female respon-

dents. However, the difference between the two sexes is not that significant. As for the educational levels of survey respondents, they exhibit a relatively even distribution among the population.

On the other hand, the respondents' religious affiliations in this survey are skewed toward Islam, with over 60 percent of participants being Muslim, followed by Buddhism (16.8%), Hinduism (15.4%), and other religions. Regarding participants' religiosity, their responses regarding religious services attendance are relatively evenly distributed across different answers, whereas most respondents reported that they pray several times a day (43.2%).

While the majority of participants display low levels of interpersonal trust (79.9%), the difference in institutional trust is not that significant between those who answered positively and those who did not (57.1% and 42.9%, respectively). Additionally, more than 75 percent of respondents expressed satisfaction with their quality of life and freedom of choice and control in their countries. They gave high ratings, ranging from 6 to 10, for their quality-of-life satisfaction and freedom of choice and control. Lastly, this paper found that most participants in the survey tend to perceive corruption as intolerable, with 67.8 percent of respondents reporting that "corruption is never justifiable". The descriptive statistics for variables without recoding are presented in Table 3.[15]

**Table 3.** Descriptive statistics.

| Variables | Min. | Mean | Max. | SD |
|---|---|---|---|---|
| Corruption Justifiability | 1 | 1.95 | 10 | 1.860 |
| Age | 1 | 1.77 | 3 | 0.670 |
| Gender | 1 | 1.45 | 2 | 0.498 |
| Marital Status | 1 | 2.32 | 6 | 2.171 |
| Education | 0 | 3.64 | 8 | 2.645 |
| Religious Denomination | 0 | 5.45 | 9 | 1.118 |
| Attendance | 1 | 3.48 | 7 | 1.874 |
| Prayer | 1 | 2.58 | 8 | 2.002 |
| Generalized Trust | 1 | 1.80 | 2 | 0.401 |
| Institutional Trust | 1 | 2.38 | 4 | 0.894 |
| Life Satisfaction | 1 | 7.34 | 10 | 2.213 |
| Freedom of Choice and Control | 1 | 7.32 | 10 | 2.285 |

Note. N = 4190; SD = standard deviation.

Table 4 displays the coefficients derived from the MLR analysis conducted on the entire sample from six South Asian countries. Firstly, the analysis compared groups with high and low justifiability of corruption and identified individual age differences, educational attainment, religious affiliation, religiosity, trust, and individualism/collectivism as significant predictors. These results indicate that two hypotheses, H2 and H3, are completely rejected in the first place.

The result for age differences reveals that individuals aged 29 or younger and those between 30 and 49 years old are more tolerant of corruption than those aged 50 or above, with statistically significant positive associations ($p < 0.001$ and $p < 0.01$, respectively). This finding can be evidence to support H1.

In addition, this paper found that individuals with moderate levels of education are more likely to consider corrupt behaviors justifiable compared to those with higher levels of education. While this finding is comparatively less significant than other variables ($p < 0.05$), it is consistent with previous literature and supports H4.

**Table 4.** An MLR analysis.

| Predictor Variables | Always Justifiable vs. Never Justifiable | | Somewhat More Justifiable vs. Never Justifiable | | Somewhat Less Justifiable vs. Never Justifiable | |
|---|---|---|---|---|---|---|
| | B | OR (95% CI) | B | OR (95% CI) | B | OR (95% CI) |
| Age (ref. age 50+) | | | | | | |
| Age up to 29 | 0.638 *** | 1.893 (1.321–2.714) | 0.571 * | 1.770 (1.098–2.854) | 0.335 | 1.397 (0.970–2.013) |
| Age 30–49 | 0.463 ** | 1.589 (1.153–2.192) | 0.506 * | 1.659 (1.082–2.545) | 0.456 ** | 1.578 (1.145–2.174) |
| Gender (ref. male) | | | | | | |
| Female | −0.069 | 0.933 (0.769–1.132) | −0.016 | 0.984 (0.755–1.282) | 0.039 | 1.040 (0.846–1.278) |
| Marital Status (ref. married) | | | | | | |
| Unmarried | 0.214 | 1.238 (0.967–1.586) | 0.144 | 1.155 (0.815–1.637) | 0.020 | 1.020 (0.772–1.347) |
| Education (ref. high education) | | | | | | |
| Low Education | 0.299 | 1.348 (0.986–1.843) | 0.705 *** | 2.024 (1.327–3.088) | 0.050 | 1.052 (0.770–1.436) |
| Medium Education | 0.389 * | 1.476 (1.093–1.994) | 0.142 | 1.153 (0.732–1.816) | −0.155 | 0.857 (0.612–1.199) |
| Religious Denominations (ref. other) | | | | | | |
| No Religion | −0.440 | 0.644 (0.228–1.819) | 0.980 | 2.663 (0.257–27.554) | −0.504 | 0.604 (0.146–2.502) |
| Roman Catholic | 0.291 | 1.338 (0.430–4.169) | 1.456 | 4.291 (0.353–52.192) | −0.790 | 0.454 (0.048–4.296) |
| Protestant | −1.470 | 0.230 (0.026–2.024) | 2.499 * | 12.176 (1.196–123.982) | 0.238 | 1.268 (0.210–7.642) |
| Muslim | −1.329 *** | 0.265 (0.126–0.558) | 0.663 | 1.941 (0.256–14.720) | 0.164 | 1.178 (0.437–3.175) |
| Hindu | −0.287 | 0.750 (0.360–1.565) | 1.404 | 4.071 (0.539–30.726) | 0.365 | 1.441 (0.540–3.849) |
| Buddhist | 0.324 | 1.382 (0.668–2.863) | 1.330 | 3.781 (0.498–28.706) | −0.107 | 0.899 (0.332–2.435) |
| Other Christian | 0.245 | 1.278 (0.336–4.853) | −17.500 | $2.511 \times 10^{-8}$ ($2.511 \times 10^{-8}$–$2.511 \times 10^{-8}$) | −18.397 | $1.024 \times 10^{-8}$ (0 [1]) |
| Religiosity | | | | | | |
| Attendance | 0.105 *** | 1.111 (1.048–1.178) | 0.107 ** | 1.113 (1.029–1.205) | −0.003 | 0.997 (0.941–1.057) |
| Prayer | −0.069 ** | 0.933 (0.886–0.984) | −0.101 ** | 0.904 (0.841–0.972) | −0.075 ** | 0.928 (0.877–0.982) |
| Trust | | | | | | |
| Generalized Trust | 0.248 * | 1.282 (1.024–1.605) | 0.120 | 1.127 (0.824–1.542) | −0.014 | 0.986 (0.767–1.266) |
| Institutional Trust | 0.022 | 1.023 (0.914–1.144) | −0.077 | 0.926 (0.801–1.071) | −0.064 | 0.938 (0.839–1.048) |
| Individualism/collectivism | | | | | | |
| Life Satisfaction | −0.026 | 0.974 (0.929–1.021) | 0.018 | 1.018 (0.954–1.088) | −0.021 | 0.979 (0.931–1.030) |
| Freedom of Choice and Control | −0.085 *** | 0.918 (0.878–0.960) | −0.119 *** | 0.888 (0.837–0.942) | −0.020 | 0.980 (0.934–1.028) |
| Intercept | −1.148 * | | −3.182 ** | | −1.321 * | |
| Chi-Square (df = 57) | 382.484 *** | | | | | |
| Pseudo R² | 0.102 [2] | | | | | |

Note. N = 4190; B = beta coefficient; OR = odds ratio; CI = confidence interval. * $p < 0.05$, ** $p < 0.01$, *** $p < 0.001$.
[1] Floating point overflow occurred while computing this statistic. Its value is therefore set to system missing.
[2] Nagelkerke Pseudo $R^2$ is presented. Cox and Snell Pseudo $R^2$ = 0.087; McFadden Pseudo $R^2$ = 0.047.

Next, an analysis reveals that the Islam variable exhibits a significant negative association with individuals' corruption justifiability, as evidenced by a significant *p*-value of < 0.001. This finding suggests that individuals who self-identify as Muslims are less likely to view corruption as justifiable, which contradicts prior research and can be evidence to reject H5.

This paper proposes one possible explanation for this unexpected negative association, which is that the relationship between an individual's religious denomination and their perception of corruption justifiability is context-dependent. More specifically, while Islam is a minority religion in most countries, it is the dominant religion in the South Asian region, given the large population of Muslims residing there. As a result of this contextual difference, South Asian Muslims' perceptions and tolerance of corruption may differ markedly from those observed at a global level.

Regarding variables of religiosity, this paper identified that the frequency of attendance at religious services is positively associated with respondents' justifiability of corruption, whereas the frequency of prayer is negatively associated with it ($p < 0.001$ and $<0.01$, respectively). These findings imply that individuals who frequently pray are less likely to tolerate corruption, whereas those who attend religious services more often are more likely to tolerate corrupt activities. Therefore, this paper can infer that both H6 and H7 are supported.

As for the unexplained positive association between the frequency of attendance at religious services and corruption justifiability (H6), this paper hypothetically argues that individuals who frequently attend religious services are more likely to be part of closed religious communities that consist of in-group members, particularly in highly religious societies. In this context, members of these communities are more likely to tolerate corrupt behaviors since such closed communities can be fertile grounds for favoritism, cronyism, and nepotism.

In the case of the variable of trust, this paper observes an unexpected positive association between people's level of generalized trust and their tolerance toward corruption ($p < 0.05$); however, the paper found that the institutional trust variable is insignificant to the outcome of interest. These findings indicate that both H8 and H9 are rejected.

The finding regarding H8 is particularly noteworthy because it suggests that South Asian respondents who have greater trust in others are more likely to justify corrupt practices, which is largely diametric to what this paper hypothesized. This remarkable finding may be due to South Asian circumstances, where corruption is prevalent and socially acceptable.

People's trust in others is based on their perception of their trustworthiness, which is contingent on the social context in which they are embedded due to the varying boundaries of social appropriateness across countries. In an environment where corruption is socially acceptable, one's ability to offer bribes or exchange favors for bribes may become another criterion for his/her trustworthiness. Consequently, people's perception of interpersonal trust may become entwined with their relaxed criteria for corruption, which could be one of the reasons for the unexpected positive association between an individual's level of generalized trust and their tolerance toward corruption in South Asia.

Lastly, this paper found that the freedom of choice and control variable is negatively related to individual corruption justifiability and is statistically significant ($p < 0.001$). This result indicates that respondents with an individualistic outlook are less likely to be tolerant of corruption, whereas people with a more collectivistic outlook are more likely to justify corruption. This finding can be partial evidence to support H10 and reject H10'.

The second column presents a comparison between respondent groups exhibiting somewhat high and low levels of corruption justifiability. The analysis indicates significant associations between corruption justifiability and variables of age, education, religious affiliation, religiosity, and individualism/collectivism. As with the first comparison, H2 and H3 are completely rejected in the first place.

Regarding age differences, this paper found a significant positive association between age groups of up to 29 and 30–49 years and corruption justifiability ($p < 0.05$). This result suggests that respondents belonging to these age groups are more likely to tolerate corruption compared to those aged 50 years and above. Similar to the results in the first column, these findings can support H1.

Next, the analysis reveals a significant positive association ($p < 0.001$) between respondents' low education levels and corruption justifiability. This finding suggests that individuals with lower educational attainment are more likely to justify corrupt behavior, which can be evidence to support H4.

As for the variables of religious denominations, this paper observed an inconsistent result that individuals who identified themselves as Protestant are more likely to see corruption as justifiable ($p < 0.05$). Similar to the previous finding regarding the Islam variable, this result can be evidence to reject H5. In this regard, this paper suggests that this outcome can also be understood through the same rationale used to explain the unexpected findings related to Islam in the first column. That is, religion may exert a varying effect on individuals' corruption justifiability depending on the context.

Concerning the association between individuals' religiosity and corruption justifiability, this paper found the result consistent with the analysis in the first column. More specifically, it indicates that individuals who attend religious services more frequently are more likely to justify corruption, while those who pray often are less likely ($p < 0.01$). As with the first comparison, these findings can be evidence to support both H6 and H7.

Similar to the comparison presented in the first column, the freedom of choice and control variable is found to be significantly and negatively associated with individual corruption justifiability ($p < 0.001$). This finding suggests that individuals with a more individualistic outlook are less likely to tolerate corruption, whereas those with a more collectivistic perspective are more prone to justify it, which can be partial evidence to support H10 and reject H10'.

In the last comparison between respondent groups of somewhat low corruption justifiability and low corruption justifiability, variables of age and prayer are found to be significantly associated with individual corruption justifiability. These findings indicate that H2, H3, H4, H5, H6, H8, H9, H10, and H10' are rejected at the very beginning of this comparison.

This paper reveals a significant positive correlation between individuals within the age range of 30–49 and their corruption justifiability in this comparison ($p < 0.01$). This finding suggests that respondents within that age group are more likely to justify corrupt behaviors compared to those aged 50 years and above, which can be evidence to support H1.

Finally, the paper observes a negative association between the variable of the frequency of prayer and corruption justifiability, suggesting that individuals who pray more frequently in South Asia are less likely to justify corruption. This relationship is significant throughout the analyses ($p < 0.01$), providing compelling evidence to support H7.

### 4.1. Robustness Check

As a robustness check, this paper additionally employs BLR instead of OPR or OLR because both the PO and the PL assumptions were rejected, as described earlier. In order to carry out BLR as a robustness check, the paper first recoded the dependent variable into a dichotomous scale with two extreme values (0 and 1), where 1 stands for "never justifiable" and 0 otherwise, following Torgler and Valev (2010), and then performed BLR. According to Table 5, some independent variables that were initially identified as statistically significant in the previous regression were found insignificant in the robustness check. Specifically, variables of religious denominations (Islam and Protestantism) and generalized trust lost significance.

Nevertheless, the outcomes obtained from the MLR analysis are deemed robust due to the significant and consistent findings in this BLR analysis for other variables such as individual age, education, religiosity, and individualism/collectivism. For example, these variables remain significant in this BLR analysis, and the coefficients for these variables are in the same direction as found in the previous analysis.

**Table 5.** A robustness check.

| Predictor Variables | B | SE | Wald | OR (95% CI) |
|---|---|---|---|---|
| Age (ref. age 50+) | | | | |
| Age up to 29 | −0.512 *** | 0.127 | 16.295 | 0.599 (0.468–0.769) |
| Age 30–49 | −0.468 *** | 0.112 | 17.415 | 0.626 (0.503–0.780) |
| Gender (ref. male) | | | | |
| Female | 0.016 | 0.072 | 0.047 | 1.016 (0.883–1.169) |
| Marital Status (ref. unmarried) | | | | |
| Married | −0.144 | 0.094 | 2.355 | 0.866 (0.720–1.041) |
| Education (ref. high education) | | | | |
| Low Education | −0.321 ** | 0.112 | 8.236 | 0.725 (0.583–0.903) |
| Medium Education | −0.156 | 0.115 | 1.823 | 0.856 (0.682–1.073) |
| Religious Denominations (ref. other) | | | | |
| No Religion | 0.294 | 0.441 | 0.446 | 1.342 (0.566–3.183) |
| Roman Catholic | −0.278 | 0.513 | 0.294 | 0.757 (0.277–2.069) |
| Protestant | −0.151 | 0.580 | 0.068 | 0.860 (0.276–2.680) |
| Muslim | 0.521 | 0.322 | 2.613 | 1.683 (0.895–3.163) |
| Hindu | −0.143 | 0.320 | 0.200 | 0.867 (0.463–1.622) |
| Buddhist | −0.342 | 0.320 | 1.145 | 0.710 (0.380–1.329) |
| Other Christian | 0.087 | 0.653 | 0.018 | 1.091 (0.303–3.921) |
| Religiosity | | | | |
| Attendance | −0.063 ** | 0.021 | 9.330 | 0.939 (0.901–0.978) |
| Prayer | 0.074 *** | 0.020 | 14.175 | 1.077 (1.036–1.120) |
| Trust | | | | |
| Generalized Trust | −0.126 | 0.085 | 2.188 | 0.882 (0.747–1.042) |
| Institutional Trust | 0.032 | 0.040 | 0.642 | 1.032 (0.955–1.115) |
| Individualism/collectivism | | | | |
| Life Satisfaction | 0.016 | 0.018 | 0.817 | 1.016 (0.982–1.052) |
| Freedom of Choice and Control | 0.068 *** | 0.017 | 16.631 | 1.070 (1.036–1.105) |
| Intercept | 0.410 | 0.373 | 1.205 | 1.507 |
| Chi-Square (df = 19) | 195.363 *** | | | |
| Pseudo $R^2$ | 0.064 [1] | | | |
| Hosmer and Lemeshow Test | 0.600 | | | |

Note. N = 4190; B = beta coefficient; SE = standard error; OR = odds ratio; CI = confidence interval. ** $p < 0.01$, *** $p < 0.001$. [1] Nagelkerke Pseudo $R^2$ is presented. Cox and Snell Pseudo $R^2$ = 0.046.

More specifically, this paper found negative correlations between variables of individuals' age differences and educational levels and respondents' justifiability of corruption and

a positive association between the variable of individuals' perceptions of freedom of choice and control and the dependent variable. These findings suggest that relatively younger and/or less educated and/or collectivistic people are more likely to justify corruption, while older and/or more educated and/or individualistic people are less likely to accept it. This finding can be robust evidence to support H1, H4, and H10.

Moreover, this paper consistently observed contrasting results regarding the variable of individual religiosity in this BLR analysis. That is, the paper found robust evidence suggesting that the more frequently people pray, the less tolerant they are of corruption, yet the more frequently people attend religious services, the more tolerant they are of corruption.

In sum, apart from a few independent variables found to be insignificant (i.e., variables of individual religious affiliations and generalized trust), the results reported in this BLR analysis are in line with the previous regression that this paper conducted.

### 4.2. Additional Analyses

While the findings obtained from the previous analyses can be considered significant and reliable, one unavoidable concern is that those results were derived from a combined dataset that was obtained from different sources and methodologies, i.e., the WVS and our own survey. Therefore, it is necessary to independently analyze two different survey data items to compare and report the main differences between them. For analysis, this paper employs an MLR model because the assumptions of PO and PL do not hold in both cases, and the preliminary Likelihood ratio test indicates the full model's significant improvement in fit over the null model in an MLR setting ($p < 0.001$). The results of these respective additional analyses are provided in Tables 6 and 7.

Let us first look at the findings obtained from the WVS survey data in Table 6. Firstly, this paper found that variables of individual religiosity, generalized trust, and individualism/collectivism are significant predictors in the first column. In fact, the pattern of results in this analysis is somewhat consistent with those found in the analysis using our combined dataset. That is, people who relatively more frequently pray and/or less frequently attend religious services and/or have lesser trust in others and/or are individualistic are less likely to tolerate corruption, while those who relatively less frequently pray and/or more frequently attend religious services and/or have greater trust and/or are collectivistic are more likely to legitimize corruption in South Asia. Some notable results regarding variables that produced effects that are diametric to conventional wisdom (i.e., variables of individual attendance at religious services and generalized trust) are consistently reported in this additional analysis as well.

The second column presents a comparison between respondent groups exhibiting somewhat high and low levels of corruption justifiability. The analysis indicates significant associations between corruption justifiability and variables of individual religious affiliation, religiosity, and individualism/collectivism. As with the first comparison and the findings derived from the analysis using our combined dataset, consistent results are found, i.e., negative connections between variables of individual prayer/freedom of choice and control and people's justifiability of corruption and positive associations between the variable of individual attendance at religious services and respondents' tolerance of corruption. One remarkable finding in this comparison is that the Islam variable exhibits a significant positive association with individuals' corruption justifiability, which is diametric to what this paper identified in the analysis using the combined dataset. Nevertheless, given its implausible values for the beta coefficient and odds ratio, this result seems to be derived from a large Muslim population in Bangladesh and Pakistan covered in the WVS and therefore to be unreliable.

In the last comparison between respondent groups of somewhat low corruption justifiability and low corruption justifiability, variables of individual age differences, educational attainments, prayer, and perception of freedom of choice and control are found to be significantly associated with respondents' corruption justifiability. While most of these findings are identified as consistent with the analyses in the first and second columns, the variable

of education shows an unexpected negative association with individual justifiability of corruption, which is diametric to what this paper found in the previous analyses. This finding suggests that individuals with moderate levels of education are less likely to consider corrupt behaviors justifiable compared to those with higher levels of education.

**Table 6.** An MLR analysis of the WVS.

| Predictor Variables | Always Justifiable vs. Never Justifiable | | Somewhat More Justifiable vs. Never Justifiable | | Somewhat Less Justifiable vs. Never Justifiable | |
|---|---|---|---|---|---|---|
| | B | OR (95% CI) | B | OR (95% CI) | B | OR (95% CI) |
| Age (ref. age 50+) | | | | | | |
| Age up to 29 | 0.341 | 1.406 (0.863–2.292) | 0.278 | 1.320 (0.776–2.245) | 0.208 | 1.232 (0.814–1.864) |
| Age 30–49 | 0.245 | 1.278 (0.818–1.995) | 0.278 | 1.321 (0.823–2.118) | 0.405 * | 1.499 (1.039–2.164) |
| Gender (ref. male) | | | | | | |
| Female | 0.097 | 1.102 (0.822–1.477) | −0.072 | 0.931 (0.668–1.296) | −0.054 | 0.948 (0.739–1.216) |
| Marital Status (ref. married) | | | | | | |
| Unmarried | −0.108 | 0.898 (0.595–1.356) | −0.268 | 0.765 (0.461–1.267) | −0.251 | 0.778 (0.533–1.135) |
| Education (ref. high education) | | | | | | |
| Low Education | 0.135 | 1.144 (0.682–1.921) | 0.330 | 1.391 (0.788–2.455) | −0.256 | 0.774 (0.535–1.121) |
| Medium Education | 0.217 | 1.242 (0.722–2.138) | −0.192 | 0.825 (0.437–1.557) | −0.459 * | 0.632 (0.420–0.951) |
| Religious Denominations (ref. other) | | | | | | |
| Muslim | −0.491 | 0.612 (0.116–3.222) | 17.443 *** | 37,605,850.090 (20,651,078.438–68,480,683.237) | 17.350 | 34,286,182.107 (0 [1]) |
| Hindu | −0.175 | 0.839 (0.143–4.939) | 18.343 | 92,496,063.913 (92,496,063.913–92,496,063.913) | 17.817 | 54,693,827.246 (0 [1]) |
| Buddhist | −17.323 | $2.998 \times 10^{-8}$ (0 [1]) | 0.609 | 1.839 (0 [1]) | 18.029 | 67,569,495.788 (0 [1]) |
| Religiosity | | | | | | |
| Attendance | 0.155 *** | 1.168 (1.073–1.272) | 0.117 * | 1.124 (1.023–1.236) | −0.023 | 0.977 (0.914–1.044) |
| Prayer | −0.291 *** | 0.748 (0.690–0.810) | −0.167 *** | 0.847 (0.769–0.933) | −0.130 *** | 0.878 (0.816–0.945) |
| Trust | | | | | | |
| Generalized Trust | 0.464 ** | 1.590 (1.150–2.199) | −0.006 | 0.994 (0.660–1.498) | 0.034 | 1.034 (0.762–1.403) |
| Institutional Trust | −0.134 | 0.874 (0.757–1.010) | −0.081 | 0.922 (0.782–1.086) | −0.085 | 0.918 (0.812–1.038) |
| Individualism/collectivism | | | | | | |
| Life Satisfaction | −0.090 ** | 0.914 (0.855–0.977) | 0.019 | 1.019 (0.941–1.103) | 0.000 | 1.000 (0.942–1.062) |
| Freedom of Choice and Control | −0.072 * | 0.930 (0.875–0.989) | −0.134 *** | 0.875 (0.818–0.935) | −0.060 * | 0.942 (0.893–0.993) |
| Intercept | 0.069 | | −18.639 *** | | −17.464 | |
| Chi-Square (df = 45) | 176.632 *** | | | | | |
| Pseudo R$^2$ | 0.075 [2] | | | | | |

Note. N = 2764; B = beta coefficient; OR = odds ratio; CI = confidence interval. * *p* < 0.05, ** *p* < 0.01, *** *p* < 0.001.
[1] Floating point overflow occurred while computing this statistic. Its value is therefore set to system missing.
[2] Nagelkerke Pseudo R$^2$ is presented. Cox and Snell Pseudo R$^2$ = 0.062; McFadden Pseudo R$^2$ = 0.037.

**Table 7.** An MLR analysis of the author's own survey.

| Predictor Variables | Always Justifiable vs. Never Justifiable | | Somewhat More Justifiable vs. Never Justifiable | | Somewhat Less Justifiable vs. Never Justifiable | |
|---|---|---|---|---|---|---|
| | B | OR (95% CI) | B | OR (95% CI) | B | OR (95% CI) |
| Age (ref. age 50+) | | | | | | |
| Age up to 29 | 0.868 ** | 2.382 (1.351–4.200) | 1.639 * | 5.152 (1.408–18.853) | 0.592 | 1.808 (0.791–4.134) |
| Age 30–49 | 0.638 * | 1.893 (1.167–3.070) | 1.472 * | 4.357 (1.310–14.492) | 0.678 | 1.970 (0.971–3.998) |
| Gender (ref. male) | | | | | | |
| Female | −0.214 | 0.807 (0.611–1.067) | −0.022 | 0.978 (0.615–1.555) | 0.094 | 1.098 (0.744–1.621) |
| Marital Status (ref. married) | | | | | | |
| Unmarried | 0.198 | 1.218 (0.846–1.756) | 0.475 | 1.608 (0.880–2.940) | 0.297 | 1.346 (0.816–2.220) |
| Education (ref. high education) | | | | | | |
| Low Education | 2.010 *** | 7.464 (2.321–24.008) | 0.597 | 1.818 (0.190–17.384) | −18.628 | $8.125 \times 10^{-9}$ ($8.125 \times 10^{-9}$–$8.125 \times 10^{-9}$) |
| Medium Education | 1.111 *** | 3.037 (1.916–4.814) | 0.570 | 1.768 (0.823–3.801) | 0.298 | 1.348 (0.652–2.785) |
| Religious Denominations (ref. other) | | | | | | |
| No Religion | −0.356 | 0.700 (0.227–2.163) | 0.325 | 1.385 (0.130–14.722) | −0.830 | 0.436 (0.101–1.877) |
| Roman Catholic | −0.047 | 0.954 (0.276–3.289) | 0.805 | 2.238 (0.177–28.265) | −1.252 | 0.286 (0.029–2.782) |
| Protestant | −2.041 | 0.130 (0.014–1.216) | 1.861 | 6.432 (0.596–69.462) | −0.254 | 0.776 (0.122–4.914) |
| Muslim | −0.128 | 0.880 (0.313–2.472) | 0.068 | 1.070 (0.088–13.081) | −0.367 | 0.693 (0.171–2.804) |
| Hindu | −0.675 | 0.509 (0.217–1.195) | 0.616 | 1.851 (0.237–14.475) | −0.104 | 0.901 (0.322–2.525) |
| Buddhist | −0.051 | 0.951 (0.411–2.200) | 0.729 | 2.072 (0.267–16.111) | −0.541 | 0.582 (0.206–1.644) |
| Other Christian | −0.118 | 0.889 (0.215–3.675) | −18.747 | $7.217 \times 10^{-9}$ ($7.217 \times 10^{-9}$–$7.217 \times 10^{-9}$) | −19.576 | $3.149 \times 10^{-9}$ ($3.149 \times 10^{-9}$–$3.149 \times 10^{-9}$) |
| Religiosity | | | | | | |
| Attendance | 0.071 | 1.073 (0.983–1.172) | 0.051 | 1.052 (0.903–1.225) | 0.010 | 1.010 (0.888–1.149) |
| Prayer | 0.055 | 1.057 (0.985–1.134) | −0.023 | 0.977 (0.872–1.095) | −0.027 | 0.974 (0.886–1.070) |
| Trust | | | | | | |
| Generalized Trust | −0.017 | 0.983 (0.703–1.374) | 0.539 * | 1.714 (1.014–2.900) | 0.087 | 1.091 (0.687–1.732) |
| Institutional Trust | 0.324 ** | 1.382 (1.138–1.679) | −0.090 | 0.914 (0.658–1.270) | −0.032 | 0.969 (0.734–1.280) |
| Individualism/collectivism | | | | | | |
| Life Satisfaction | 0.046 | 1.047 (0.972–1.129) | 0.023 | 1.023 (0.904–1.158) | −0.079 | 0.924 (0.836–1.021) |
| Freedom of Choice and Control | −0.092 * | 0.912 (0.845–0.984) | −0.083 | 0.921 (0.812–1.044) | 0.100 | 1.105 (0.993–1.229) |
| Intercept | −2.334 *** | | −4.659 *** | | −2.283 ** | |
| Chi-Square (df = 57) | 175.071 *** | | | | | |
| Pseudo R$^2$ | 0.131 [1] | | | | | |

Note. N = 1426; B = beta coefficient; OR = odds ratio; CI = confidence interval. * $p < 0.05$, ** $p < 0.01$, *** $p < 0.001$.
[1] Nagelkerke Pseudo R$^2$ is presented. Cox and Snell Pseudo R$^2$ = 0.116; McFadden Pseudo R$^2$ = 0.057.

This paper interprets this outstanding finding as a result of both countries' education curricula building a corruption culture through wrong messages (Sakib 2019). That is, an education curriculum in both countries is less likely to provide moral education against corruption. Instead, it focuses more on making a profit even if a means for it is normatively

wrong, thereby promoting a corruption-friendly culture. Consequently, the more people in both countries are educated, the more likely they are to be exposed to excessive materialistic values through incorrect messages in the curriculum; therefore, it may produce this remarkable finding.

However, we should be cautious about this interpretation for the following reasons. First, the interpretation is not fully corroborated by empirical evidence. Second, only one coefficient on the variable associated with education is statistically significant at the 10 percent level, which may not necessarily represent a meaningful difference.

Subsequently, this paper reports the results derived from the analysis using the author's own survey data targeting respondents from Bhutan, India, Nepal, and Sri Lanka in Table 7. The first column presents a comparison between participant groups showing high and low levels of corruption justifiability. The analysis reports that variables of individuals' age differences, education, institutional trust, and perceptions of freedom of choice and control have significant effects on respondents' justifiability of corruption. Apart from the consistent results, one notable finding is that the variable of individuals' institutional trust previously identified as insignificant is found to be significantly positively correlated to the dependent variable. This result is remarkable because it indicates that the higher trust people possess in institutions and civil services in their countries, the more likely they are to tolerate corruption.

Regarding this notable finding, this paper suggests that this result can also be understood through the same logic used to explain the unexpected finding related to generalized trust found in the analysis using the combined dataset. Not only people's trust in others but their trust in institutions and civil services may also be contingent on their perception of trustworthiness. As this perception can vary across the contexts in which they are embedded, people's perception of institutional trust may be intertwined with their more lenient standards for corruption in relatively more corrupt settings. Considering that a majority of the countries examined in our own survey are deemed comparatively more corrupt on a global scale, the aforementioned reasoning may account for an unexpected positive correlation between individuals' institutional trust and their tolerance toward corruption. However, since this interpretation has not been empirically proved, and Bhutan, one of the countries covered in this paper, has rather been classified as one of the most transparent countries in the world (TI 2023), future research needs to empirically test this hypothesis.

In addition, variables of age and generalized trust are significantly correlated to individual corruption tolerance in the second column, whereas no variables are found to be significant to the dependent variable in the third column. The results identified in the second comparison are found to be consistent with preceding analyses in that the coefficients for those variables work the same way as identified in the previous regression analysis. As in other analyses using the combined dataset and the WVS, a generalized trust variable is found to be positively related to the dependent variable. Regarding this finding, this paper posits that it results from those countries' circumstances in which corruption is prevalent and socially acceptable. However, additional research is needed to confirm our hypothesis, as described earlier.

When comparing the outcomes obtained from the WVS and our own survey, this paper identified significant disparities, particularly in relation to the statistical significance of variables associated with age and education. It was observed that in the analysis conducted using the WVS dataset, only one coefficient on each of these respective variables was statistically significant at the 10 percent level. However, when our own survey dataset was considered, multiple coefficients demonstrated meaningful significance.

There might be several reasons why the statistical significance of the same variables varied across the analyses. One of the plausible reasons is that this might be the case for different data elicitation methods or variations in respondent characteristics between the two surveys. For instance, the educational attainments of respondents in the WVS dataset are skewed toward lower levels, with 61.3 percent of participants possessing a low educational level. In contrast, over 90 percent of our own survey sample had a high

educational level (90.9%). Such discrepancies in the educational profiles of the samples may contribute to the observed disparities in statistical significance.

## 5. Discussion and Conclusion

Corruption is a grave issue in the South Asian region. Hence, many political leaders and policy makers in the region have unanimously raised their voices against it (UNODC 2018) and focused on identifying several factors that affect corruption. Despite the efforts to tackle corruption, its individual-level determinants have not been fully identified. Further, to the best of the author's knowledge, no previous research has investigated the impact of individual-level variables on corruption in the South Asian context. With this background, this paper sought to fill this gap by examining the relationship between selected individual variables and corruption justifiability in South Asia.

This paper carried out MLR for analysis and identified that individual age differences, educational levels, religious denominations (particularly Protestantism and Islam), religiosity, generalized trust, and individualism/collectivism (as measured by freedom of choice and control) have significant associations with respondents' tolerance toward corruption. However, in a further BLR analysis as a robustness check, variables of individual religious affiliations and generalized trust turned out to be insignificant to the outcome of interest.

Nevertheless, this paper observed that apart from those variables found to be insignificant in the BLR analysis, other variables of individual age, education, religiosity, and individualism/collectivism remained robust, and the coefficients for those variables work in the same way as identified in the previous regression analysis. These findings can support the robustness of the findings obtained from the MLR analysis by suggesting that relatively younger and/or less educated and/or collectivistic people are more likely to justify corruption, while older and/or more educated and/or individualistic people are less likely to legitimize it.

Meanwhile, what should be noted here is that variables of individual religiosity were found to be robust and produce effects that are in line with the findings identified in the existing literature but are also diametric to conventional wisdom. More specifically, the results indicate that the more frequently people pray, the less tolerant they are of corruption, as expected, yet the more frequently people attend religious services, the more tolerant they are of corruption.

Regarding this remarkable finding, this paper hypothetically claims that the more people participate in religious services, the more likely they are to be influenced by a religious community, which is likely a closed circle of in-group members and therefore can be a fertile ground for corruption, especially in a highly religious society such as South Asia. However, future research is needed to confirm this hypothesis.

In addition, this paper performed additional analyses separately using the WVS and its own survey data to report the main differences. This supplementary work is profoundly crucial because the paper combined different observations from two surveys into a single dataset and utilized them (Rezaei Ghahroodi 2023). Therefore, even though we justified the data integration based on the relevant existing literature, it is imperative to consider those two samples independently in order to identify whether there exists a significant difference between them.

In the analyses, variables of individuals' age, education, religiosity, trust, and individualism/collectivism were found to be significant to respondents' justifiability of corruption, and the coefficients for most variables worked the same way as identified in the previous regression analysis using the combined dataset.

However, this paper found the following notable results. Firstly, in the analysis using the WVS dataset, an individual education variable was somehow found to be significantly negatively correlated to the dependent variable. This result indicates that individuals with moderate levels of education are less likely to consider corrupt behaviors justifiable compared to those with higher levels of education, especially in Bangladesh and Pakistan, which is diametric to what was found in previous analyses.

This paper presented a plausible hypothesis about this unexpected negative correlation, arguing that it may be due to both countries' problematic education promoting a corruption-friendly culture through wrong messages (Sakib 2019). However, since only one coefficient on the education variable was statistically significant at the 10 percent level, this finding may not necessarily represent a meaningful difference.

On the other hand, the paper found a positive correlation between the variable of individuals' generalized trust and their tolerance toward corruption in the analysis using our own survey dataset. This paper interpreted this notable finding as a result of people's perception of interpersonal trust varying across the contexts, where they are embedded.

Lastly, this paper observed that several coefficients associated with the age and education variables were meaningfully significant in the analysis using the WVS, whereas only one coefficient for each of these variables was statistically significant at the 10 percent level in the analysis using our own survey dataset.

On this discrepancy, this paper argued that this might be the case for different data elicitation methods or different respondent characteristics in two surveys, which can lead us to the conclusion that more caution is necessary when interpreting the findings obtained from our main analysis using the combined dataset.

In light of the findings, this paper may present the following contributions. Firstly, even though most authors have conducted corruption research on a global level or have conducted case studies, they have not focused on the individual-level determinants of corruption within the South Asian region. In this context, this paper may shed light on the effects of individual factors on people's corruption justifiability in South Asia, where much of scholars in the corruption-related discipline have paid less attention.

Secondly, this paper replicated the WVS questionnaire, translated it into the respective countries' official languages, and further conducted its own survey research to overcome data constraints. This additional data collection not only improves the originality and reliability of this paper but also provides a comprehensive dataset for future research.

Meanwhile, the results identified in this paper have some policy implications. The paper found that active participation in religious services does not help to reduce corruption at the individual level but rather promotes it. Based on this result, we could jump to the hasty conclusion that decreasing the frequency of people's participation in religious services may prevent South Asian people from engaging in corrupt behaviors. However, such a recommendation should be treated with caution because stopping people with religion from attending religious services is, however well-meaning in intent, religious persecution, and the effect of religiosity on corruption may not be direct but possibly mediated by other essential factors (Flavin and Ledet 2013).

With this background, this paper alternatively suggests that South Asian governments and decision makers should aim to tackle a pervasive culture of corruption among religious communities. More specifically, as in the Kenya case (Ethics and Anti-Corruption Commission 2022), the paper calls for nationwide dissemination of anti-corruption training materials aimed at mainstreaming integrity and an anti-corruption culture among religious communities in South Asia. Here, those materials against corruption should be developed through a partnership with independent oversight institutions and civil society organizations to minimize political interference by powerful religious leaders who are amassing wealth in that corrupt environment. The training material should not be limited to booklets, brochures, posters, and so on, but should encompass activities-related and interactive content, such as participation in events with anti-corruption content and so forth. This suggestion may help to carry anti-corruption messages and practical tips for religious leaders and followers, thus minimizing the possibility that religious communities in South Asia would be fertile ground for corruption.

Despite all the aforementioned contributions and implications, this paper is subject to the following limitations. Firstly, this paper supplemented the insufficient data through online survey software for some unavoidable reasons. While this additional data gathering helped enhance the originality and reliability of this paper, the web-based survey technique

may easily suffer from a number of methodological limitations, such as selection bias and the lack of control over the sample size.

Secondly, the participants in our survey were not representative of the population due to data constraints, and the sample size varied across countries. Thus, the results reported in this paper are difficult to use to extrapolate for the whole population residing in South Asia.

With this background, to ensure the generalizability of the findings to the broader South Asian population, including Afghanistan and the Maldives, which were excluded from this paper, future research should aim to collect sufficient data from a larger group that accurately represents the characteristics of a larger population through the use of traditional face-to-face surveys. Prospective research could also carry out empirical analysis with those data in order to identify whether our findings are still applicable.

**Funding:** This research received no external funding.

**Informed Consent Statement:** Not applicable.

**Data Availability Statement:** All data are available from the author upon request.

**Acknowledgments:** This paper draws upon the fourth chapter of the author's doctoral dissertation titled "Public Sector Corruption in South Asia 2006–2022: Determinants and Policy Implications".

**Conflicts of Interest:** The author declares no conflict of interest.

## Notes

1    While the definition of corruption has been subject to multiple interpretations, the purpose of this paper is not to engage in such debates. Thus, to ensure rigorous empirical research, this paper adopts the most agreed-on and common definition when defining the concept of corruption, which is "the misuse of public office for private gain" (World Bank 1997). For more detailed definitional debates over corruption, see Heidenheimer (1970).

2    The CPI is a composite indicator to measure levels of public sector corruption on a scale of 0 (very corrupt) to 100 (very clean).

3    It is important to note here that corruption has been known to have both positive and negative sides (e.g., Leitão 2021; Dang et al. 2022; Han 2022; Nguyen 2022; Almustafa et al. 2023; Han 2023). Nevertheless, corruption, by and large, is considered to have detrimental effects on political, economic, and social areas (Méon and Sekkat 2005). Against this backdrop, the negative implications of corruption are emphasized in this paper.

4    This paper confines its spatial scope to six South Asian countries, namely, Bangladesh, Bhutan, India, Nepal, Pakistan, and Sri Lanka, where all the data are available.

5    In fact, all the previous studies surveyed in this paper have employed the measure of corruption justifiability as a proxy for assessing corruption levels at the individual level.

6    Establishing causality between individuals' religious affiliations and their tolerance toward corruption requires careful examination; otherwise, it may lead to a premature inference that "people with particular religious denomination tend to perceive corruption as more justifiable". Therefore, this paper notes that our individual-level analysis results for the connection between them are not definitive but indicative of a causal link between them.

7    Uslaner's first draft was presented at the Conference on Political Scandals, Past, and Present at the University of Salford in 2001.

8    Although the item regarding religious affiliations from the WVS has significant advantages, one of its main limitations is that it does not consider various sects and attributes within the same religion. For instance, although there are two main sects in Islam, Sunni and Shia, the WVS does not differentiate Muslims according to their diverse religious factions. This deficiency in the survey may result in biased outcomes for the religious denomination variable since the variation between the two sects may be as significant as the disparity between Hindus and Muslims. Nevertheless, this paper opts to utilize the WVS dataset because there are no comparable datasets capturing all the different sects and characteristics within the religions, and the WVS is widely regarded as a valid and reliable measure of religious affiliations.

9    One may raise the question of whether the two items about individuals' attendance at religious services and prayer have mutually exclusive and exhaustive categories since individuals can pray by attending religious services. However, the two items can be differentiated by offering respondents the option to select "pray only when attending religious services."

10   The cultural dimension of individualism/collectivism is generally measured by using the cultural dimension index created by Hofstede. However, this index is not suitable for our individual-level independent variables, as it measures values for multiple cultural dimensions at the country level. To address this issue, Kang and Kwon (2018) propose using two items about respondents' life satisfaction and perception of freedom of choice and control from the WVS as alternatives, as they have a strong correlation

with Hofstede's individualism/collectivism dimension index. Against this theoretical backdrop, this paper uses these two variables to measure the cultural dimension at the individual level.

[11] The survey questionnaire was translated into Dzongkha, Hindi, Nepali, and Sinhala for Bhutanese, Indian, Nepalese, and Sri Lankan respondents, respectively. In the case of the questionnaire for Sri Lanka, where more than one local language, Sinhala and Tamil, are accorded official status by the government, this paper translated it only into Sinhala, considering the relatively larger percentage of Sinhala-speaking individuals in the country. The English and local-language questionnaires can be obtained upon request.

[12] Native speakers translated the questionnaire into Hindi, Nepali, and Sinhala, whereas an expert with country-specific knowledge translated it into Dzongkha. Personal information is available upon request.

[13] The initial sample sizes for Bangladesh and Pakistan were 1200 and 1995, respectively. However, after the exclusion of non-response and "don't know" answers, the sample sizes for Bangladesh and Pakistan were reduced to 1109 and 1655, respectively.

[14] Before carrying out the Likelihood ratio test, this paper first conducted both Pearson's chi-square and the Deviance chi-square tests to assess the goodness-of-fit of the MLR model. In these tests, Pearson's chi-square test indicates poor model fit, whereas the Deviance chi-square test exhibits a good fit to the data. Even though both tests do not always necessarily agree, neither the chi-square tests of Pearson nor Deviance can be considered reliable tests for goodness of fit in this case.

[15] Note that Variance Inflation Factors (VIFs) are all less than 10 in all models across countries. Therefore, multicollinearity appears not to be a problem in the subsequent analysis. The VIF results will be available upon request.

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
