# Peer review of "Examining Determinants of Corruption at the Individual Level in South Asia"

_economies, doi:10.3390/economies11070179_

Round 1

Reviewer 1 Report (Previous Reviewer 2)

Examining Determinants of Corruption at the Individual Level in South Asia

I see that this paper has been reviewed by me before, after rejecting and resubmit now. I looked through the response letter and found that the authors replied that they had edited it according to my request. However, at a glance, the author has not yet fully edited it. For example, the studies I proposed were not fully reviewed and cited, Nguyen, Q. K. (2022), Nguyen and Dang (2022). I still haven't seen the author cite, the authors need to update these studies.

Therefore, I resubmit my previous request, the authors review the comments again and fully supplement them. I will review it after the author has finished editing.

Nguyen, Q. K. (2022). Audit committee structure, institutional quality, and bank stability: evidence from ASEAN countries. Finance Research Letters, 46, 102369. 

Nguyen, Q. K., & Dang, V. C. (2022). Does the country’s institutional quality enhance the role of risk governance in preventing bank risk? Applied Economics Letters, 1-4.

Previous comment:

-In the introduction, the authors argue that "Even though certain factors are revealed as significant globally or elsewhere, their effects would vary depending on the regional context". However, the authors do not show that South Asia has anything special to study. In the first paragraph, the author thinks that "most South Asian countries have ranked among the most corrupt countries in the world for at least a decade", this statement is too general, specifically which country, how many countries, and how ratings?

- The literature review section has not improved much, the studies I propose have not been fully reviewed and cited (only 2/4).

- In conclusion, I asked the authors to make a brief presentation but nothing seems to improve. In this part, the author seems to re-analyze the research results. With 2 pages of content like that can't be the conclusion.

English is quite good.

Author Response

Reviewer 2 Report (Previous Reviewer 1)

This work is interesting, but my previous concerns have not been fully addressed.

1)The authors now report probit estimates to test robustness, which reduces my first concern (albeit I would have preferred the binomial model as the main model, as in Torgler and Valev, 2010).

The authors state on pages 10 and 15 of the revised version of the paper, as they did in the initial version, that the “previously employed ordinal logistic regression model failed to pass a test of parallel lines”. Please share the results of your testing on previously ordered logit/probit estimates. This is important since an ordered model is the best choice for your dependent variable whereas an MLR is usually used when dealing with categorical outcomes.  Please note that in my last report I wrote: “You might also provide further statistical analysis to support your final decision (for instance, you chose an MLR even though you didn't provide any test results)”.

2) In my previous two reports, I highlighted an additional issue: you combine observations from two surveys, the World Value Survey and a survey conducted using an online survey tool provided by Google, into a single dataset.

On page 9, you write that “ In this survey research, this paper replicates the WVS’s original questionnaire written in English for comparability and translates it into the respective countries’ official languages, following Han (2022)” . Unfortunately, I am not able to read Han (2022) since it is a doctoral thesis. Your response, however, does not address my worry about the merging of two datasets. I believe the topic is even more difficult (see for example http://nap.nationalacademies.org/24893, or https://doi.org/10.1007/s10260-023-00693-2). Anyway, for the sake of brevity, I believe that you should ALSO consider the two samples independently in order to report main differences. Furthermore, estimates based on World Value Survey data might also contain survey weights

Minor editing of English language required. For example, the phrase "estimated ordinal logistic" seems to be preferable to "employed ordinal logistic."

Round 2

Reviewer 1 Report (Previous Reviewer 2)

This version is better and can be published

English is ok

Author Response

Reviewer 2 Report (Previous Reviewer 1)

In my opinion, the biggest sticking points have been resolved. My last concern is about your statements in the conclusions.

i) You say:  “Firstly, in the analysis using the WVS dataset, an individual education variable  was somehow found to be significantly positively correlated to the dependent variable. This result indicates that the relatively less educated people especially in Bangladesh and  Pakistan are less likely to tolerate corruption, which is diametric to what this paper found  in previous analyses. “   

Are you sure? Only one coefficient on the dummy variables associated to education is statistically significant at the 10% level in Table 6, and this, in my opinion, does not represent a meaningful difference.

ii) In the conclusions you say  : “On the other hand, the paper found a positive correlation between a variable on individuals’ institutional trust and their tolerance toward corruption in the analysis using …”.

I don't think this distinction is crucial because you obtain a positive and significant result on "generalized trust" and the two variables are likely correlated.

iii) I do not understand your final consideration: “In addition, all the selected independent variables were found insignificant to the dependent variable in a comparison between respondent groups exhibiting somewhat  less and low levels of corruption justifiability in this analysis….”.

What do you mean?

Overall, I see that the biggest disparities between the WVS and your survey results are in education and age, which are not significant when you consider the WVS dataset . This could be due to different data elicitation methods or different respondent characteristics - For example, did you compare the two samples' mean age and education (as well as standard deviation)?  

However, the fundamental outcomes on "attendance," "prayer," "trust”, and "freedom of choice" remain...Is this correct?  

Section 4.2 (including Tables 6 and 7)  can be placed in the Appendix if you like, with the  main results reported in the text.

Minor points

1)Make a note of the following sentence: “The PO assumption is  violated, as evidenced by a Likelihood ratio chi-square 164.878 with a p-value less than 565 0.05 (p < 0.001), and the PL assumption is violated as well with a Likelihood ratio chi-  square 219.456 with a p-value less than 0.05 (p < 0.001). Typically, a p-value should exceed a significance level of 0.05 to support the validity of the assumptions. As one of the key assumptions underlying ordered logistic (and ordered probit) regression is that the relationship between each pair of outcome groups is the same, we would need a different model in this case”.   

Moreover, It is preferable to say "do not hold" rather than "are violated."

2)Make a note of the following sentence :” Before employing the MLR, however, it is preliminarily imperative to conduct a goodness-of-fit test in order to identify whether the MLR model exhibits a good fit for this paper. In this test,while Pearson’s chi-square test indicates that the model does not fit our  data well, the Deviance chi-square test exhibits a good fit to the data. Even though both tests do not always necessarily agree, neither of the chi-square tests of Pearson and Deviance can be considered to provide a reliable test for goodness of fit in this case”.

3)Even if the BLR are reported to check for robustness, they should not be called a "robustness test." In my opinion, sentences as  “The  results of the robustness test are shown in Table  5” ( pag 16)….”the results reported in this  further robustness test” (p. 16) should be removed

 Additional editing work is required.

For example, check the following  sentences: 

 “Apart from those variables, nevertheless, other variables on individual age, education, religiosity, and individualism/collectivism are found to be robust” Pag. 16…… It is worth noting that the outcomes (rather than the variables) are robust.

“More specifically, this paper found negative correlations between variables on individuals’ age  differences and educational levels and their justifiability of corruption and a positive association between a variable on individuals’ perceptions of freedom of choice and control  and the dependent variable  “….  Their justifiability, what do you mean?   ….the variable on individual perception…

“a variable on individuals’ institutional trust and their tolerance toward corruption”  …their?

“this paper decides”.

“Non married”  

“Before employing the MLR, however, it is preliminarily imperative to conduct a goodness-of-fit test in order to identify whether the MLR model exhibits a good fit for this paper.”

Author Response

This manuscript is a resubmission of an earlier submission. The following is a list of the peer review reports and author responses from that submission.

Round 1

Reviewer 1 Report

The investigation into people's perceptions of corruption in South Asian regions makes the work interesting. Since The World Value Survey dataset does not contain observations from 4 South Asian regions, the authors conducted a survey to obtain responses from these countries.

The work should be enhanced, though.

1)The World Value Survey observations and the authors' own survey are combined into one dataset.  I believe the authors ought to examine the two datasets independently as well.

2)Multinomial Logit Regressions are the foundation of the empirical analysis. I think the authors should include additional estimates based on ordered logit or probit models since the dependent variable is ordinal. In order to determine if a certain individual's opinion is above or below a threshold level (e.g. the median value), the authors may additionally use a binomial logit or probit model as a robustness check.

3)The review of the literature is excessively long. The writers should also include a section with their research hypotheses and/or the anticipated results based on the literature review.

4)The authors should add a section describing the econometric model and the variables. The statistics should be given independently from the variable descriptions. The descriptive statistics table is overly lengthy and difficult to understand.

Quality of English Language: Moderate editing of English Language

Reviewer 2 Report

Article title:  Examining Determinants of Corruption at the Individual Level in South Asia

After reviewing this paper carefully, I have some comments below: 

- In the introduction, the authors need to introduce research gaps, research questions, and the contribution of this study. I didn’t see them in the current version. 

- Corruption has both positive and negative sides that have been found in many previous studies as part of institutional quality. The authors should highlight the motivation of this research in the introduction by reviewing these studies. I suggest the author review and cite some recent studies such as Almustafa et al. (2023); Dang et al. (2022); Leitão, N. C. (2021); Nguyen (2022) … (see reference)

- Authors need to add theory background in Section 2. The authors need to make a specific hypothesis for each factor in this section.

- Section 3 should be Data and Method. Besides the data, the authors also need to clearly introduce the research methodology applied in this paper.

In section 4, the authors need to analyze the results in the direction of supporting or rejecting the hypotheses that are given in section 2. How do the results of this study support the theories or previous studies?

- The analysis of research results must be in section 4. Section 5 should briefly present research results, implications, limitations and suggest directions for further research.

- There are some grammatical errors, which the authors need to check carefully

Overall, this study is not yet up to publication standard, so the authors should review and correct all (noticeably all) of my comments above and should be reconsidered before publication.

References

Almustafa, H., et al. (2023). The impact of COVID-19 on firm risk and performance in MENA countries: Does national governance quality matter? PloS one, 18(2), e0281148. 

Dang, V. C., et al (2022). Corruption, institutional quality and shadow economy in Asian countries. Applied Economics Letters, 1-6. 

Leitão, N. C. (2021). The effects of corruption, renewable energy, trade and CO2 emissions. Economies, 9(2), 62.

Nguyen, Q. K. (2022). Audit committee structure, institutional quality, and bank stability: evidence from ASEAN countries. Finance Research Letters, 46, 102369. 

The quality is not so good.

Round 2

Reviewer 1 Report

This work is interesting but my earlier worries are still present (see may previous review, points 1 and 2).

In particular:

1) The authors claim that they cannot employ a binomial probit since their dependant variable has a range of 0-3. Please note that the original variable utilized in this study as the dependent variable, taken from the WVS, is not bivariate but instead ranges from 0 to 10, with the two extreme values "never justified" and "always justified." As a result, it should the outcome variable in an ordered probit/logit model.  I suggested using a probit or binomial logit model as a robustness check to assess whether a specific person's opinion is above or below a certain level (such as the median value).

In fact, Torgler and Valev (2010) – see your references- recoded the ten-scale index with the two extreme points "never justified" and "always justified" into a two-point scale (0, 1), with the number 1 denoting "never justifiable" . Consistently with the transformed variable, they estimate a binomial probit. Then they show that their results are robust when changing the structure of the dependent variable (e.g., values between 0 and 3) and the model (ordered probit,  not a MLR).

Similar to Torgler and Valev (2010), you modified the dependent variable's structure (e.g., you now have values between 0 and 3) and used an MLR. This is  an unusual choice because multinomial logistic is typically used when dealing with categorical outcomes. Like in Torgler and Valev (2010), I proposed employing additionally a binomial probit or an ordered probit to test robustness. You might also provide further statistical analysis to support your final decision (for instance, you chose an MLR even though you didn't provide any test results).

2) The specific issue is that you rely on datasets that were obtained from various surveys that were conducted using various methodologies. Could you give examples of earlier works that used the similar methodology?  In my opinion, you should also take the two samples into independent consideration.

Minor points:

1) Table 1 is not informative since you basically repeat what you mentioned in the first column in the second column.   Please, see Table A.1 in Torgler and Valev (2010) for an example.

2) Table 2 is lengthy and challenging for a number of reasons: a) when using ordinal variables, you should report mean values rather than the number of observations for each value; b) since Table 1 contains this information, it is useless to give the minimum and maximum values for each variable ; c) finally, in table 2, you might just specify the variable that was employed in the empirical analysis, such as "female." Overall, see  Table A.2 in Torgler and Valev (2010) as an example.

3) The description of the econometric model contains unecessary claims, for example: ”Logistic regression also  called a logit model”…

Moderate editing of English Language 

Reviewer 2 Report

The authors have made some minor corrections but have not yet met my requirements.

In the introduction, the authors argue that "Even though certain factors are revealed as significant globally or elsewhere, their effects would vary depending on the regional context". However, the authors do not show that South Asia has anything special to study. In the first paragraph, the author thinks that "most South Asian countries have ranked among the most corrupt countries in the world for at least a decade", this statement is too general, specifically which country, how many countries, and how ratings?

- The literature review section has not improved much, the studies I propose have not been fully reviewed and cited (only 2/4).

- In conclusion, I asked the authors to make a brief presentation but nothing seems to improve. In this part, the author seems to re-analyze the research results. With 2 pages of content like that can't be the conclusion.

The authors should fully implement the above comments.

It is ok